# Do Anti-Phage Antibodies Persist after Phage Therapy? A Preliminary Report

**DOI:** 10.3390/antibiotics11101358

**Published:** 2022-10-05

**Authors:** Marzanna Łusiak-Szelachowska, Ryszard Międzybrodzki, Paweł Rogóż, Beata Weber-Dąbrowska, Maciej Żaczek, Andrzej Górski

**Affiliations:** 1Bacteriophage Laboratory, Hirszfeld Institute of Immunology and Experimental Therapy, Polish Academy of Sciences (HIIET PAS), 53-114 Wrocław, Poland; 2Phage Therapy Unit, Hirszfeld Institute of Immunology and Experimental Therapy, Polish Academy of Sciences (HIIET PAS), 53-114 Wrocław, Poland; 3Department of Clinical Immunology, Transplantation Institute, Medical University of Warsaw, 02-006 Warsaw, Poland; 4Infant Jesus Hospital, Medical University of Warsaw, 02-005 Warsaw, Poland

**Keywords:** anti-phage antibody, immune system, phage neutralization, phage therapy

## Abstract

Phages are immunogenic and may evoke an immune response following their administration. Consequently, patients undergoing phage therapy (PT) produce phage-neutralizing serum antibodies. The clinical significance of this phenomenon for the success or failure of the therapy is currently unclear. Interestingly, even a strong anti-phage humoral response does not exclude the success of PT. On the other hand, it cannot be ruled out that phage–antibody complexes may be trapped in tissues and organs causing injury and late complications of PT. Therefore, patients should be monitored for the presence of serum antibodies and therapy discontinued if their level is high. Our preliminary data suggest that the kinetics of the disappearance of those antibodies may vary from patient to patient and in some cases may take more than a year.

## 1. Introduction

Phage therapy (PT) uses natural predators of bacteria—bacteriophages (phages)—to eliminate antibiotic-resistant pathogens and is currently one of the most promising alternatives to antibiotics in the era of increasing multidrug resistance of bacteria. Accumulating data suggest its efficacy and safety in animal models of bacterial infections as well in human clinics. Although no clinical trial has been successfully completed to confirm the efficacy of PT in accord with the current standards of evidence-based medicine, several more trials are currently underway. As recently pointed out, PT has the potential to enhance antibiotic efficacy, protect newly developed antibiotics, and provide a last resort in cases of complete antibiotic failure [1].

Phages are antigens and interact with the immune system. The current knowledge on immune responses elicited by phages and their potential effects has been recently outlined [2].

Accumulating data indicate that phages may elicit strong humoral responses in both animals and humans [3,4,5,6]. Our earlier data revealed that patients on PT produce serum anti-phage antibodies; however, their clinical significance is unclear as they may mark both a positive and negative prognosis [5,6]. Preliminary data suggested that induction of antiphage activity of sera (AAS) during or after local or local/oral PT does not exclude a positive PT result [5]. High AAS was showed in seven cases with a positive response to PT, whereas there was an inadequate response in eight others with high AAS. The next article confirmed that the level of AAS in patients treated with phages is not correlated with the outcome of PT [6]. Humoral immune response to PT depends on many factors including duration of treatment, phage immunogenicity, phage dosage, and route of administration, as well as the immune status of the patient [7]. In fact, in some cases, as much as 6 months of PT is needed to induce significant humoral responses [8]. However, the question of antibody persistence after termination of the therapy has not been reported thus far. In this report we present preliminary data addressing this important issue. 

## 2. Materials and Methods

### 2.1. Patients

Four patients with bacterial bone infection described in this article (Table 1) underwent PT between 2014 and 2021 at Hirszfeld Institute of Immunology and Experimental Therapy (HIIET) Phage Therapy Unit (PTU). Patients no. 1, 3 and 4 used monovalent phage lysate *Staphylococcus aureus* Staph_1N, *Pseudomonas aeruginosa* Ps_2N or *Escherichia coli* Ecol_L-4 (Table 1). Patient no. 2 used a lysate of the *S. aureus* MS-1 phage cocktail in the first PT cycle, a lysate of the monovalent *S. aureus* Staph_676Z phage in the second PT cycle and a lysate of the monovalent *S. aureus* Staph_1N phage in the third PT cycle. All phage preparations had concentrations of 10^6^–10^8^ plaque forming unit/mL (pfu/mL). Phages were selected for treatment on the basis of phage typing and were used with lytic activity against the patient’s bacterial strain either locally or both locally and orally according to the protocol “Experimental phage therapy of drug-resistant bacterial infections, including MRSA infections” [9]. Blood was collected before PT, during PT, and after PT. The blood was centrifuged at 1500× *g* for 10 min and sera were stored at −70 °C. AAS was tested immediately after obtaining sera. The neutralization of phages by patients’ sera was performed after obtaining the consent of the Bioethics Committee of the Wrocław Medical University (Poland). All patients gave written informed consent.

### 2.2. Plate Phage Neutralization Test

The level of AAS of patients undergoing PT was tested with the plate phage neutralization test as described earlier [6]. Briefly, the serum was diluted from 1:10 up to 1:1500 and the phage at a concentration of 1 × 10^6^ pfu/mL (50 μL) was mixed with diluted serum (450 μL). The mixture was incubated at 37 °C for 30 min. After incubation, the mixture was diluted 100-fold with cold-enriched broth. The phage titer was determined by the double-agar layer method described by Adams [10]. AAS was assessed as the rate of phage inactivation K (K = 2.3 × (D/T) × log (P0/Pt), where D is the reciprocal of the serum dilution, T is the phage-serum reaction time (30 min.), P0 is the phage titer at the beginning of the phage-serum reaction, and Pt is the phage titer after time T of the phage-serum reaction). The variability between the individual values did not exceed 30.7%. A K rate of less than 5 was determined to be a low level of phage inactivation, a K of between 5 and 18 as a medium level of phage inactivation, and above 18 as a high level of phage inactivation.

### 2.3. Categories of the Results of PT

The outcome of PT was assessed according to Międzybrodzki et al. [9]. 

Categories A–C were described as positive responses to PT:

A—pathogen eradication and/or recovery (eradication confirmed by the results of bacterial cultures; recovery refers to wound healing or complete subsidence of the infection symptoms); 

B—good clinical result (almost complete subsidence of some infection symptoms, together with significant improvement in the patient’s general condition after completion of PT); 

C—clinical improvement (discernible reduction in the intensity of some infection symptoms after completion of PT to a degree not observed before PT, when no treatment was used).

Categories D–G were described as inadequate responses to PT:

D—questionable clinical improvement (reduction in the intensity of some infection symptoms to a degree that could also be observed before PT);

E—transient clinical improvement (reduction in the intensity of some infection symptoms observed only during application of phage preparations and not after termination of PT);

F—no response to treatment (lack of reduction in the intensity of some infection symptoms observed before PT);

G—clinical deterioration (exacerbation of symptoms of infection at the end of PT).

## 3. Results

As mentioned earlier [7], patients were treated individually according to the protocol of the experimental phage therapy and visits varied from one to eleven depending on the course of the therapy. It was impossible to standardize sample collections similarly to protocols of clinical trials.

Four patients with bone infection were treated either locally or both locally and orally with phages (Table 1). Patients no. 1, 3 and 4 had one PT cycle. Patient no. 2 had three PT cycles with different types of staphylococcal phages described in the Materials and Methods section. Prior to the first PT cycle, patient no. 2 had a low rate of phage inactivation K = 0.01. During the first, second and third PT cycle, he had a high level of K (20.76, 42.32 and 133.90, respectively). The antiphage activity of sera of four patients treated with phages is presented in Figure 1. Prior to PT, patients no. 1–4 had low AAS levels (K = 0–0.02). The data from four patients indicated that the K rate increased during PT or shortly after PT and declined long after treatment, e.g., from 5 months to 1 year and 8 months after PT. Patients no. 1 and 2 applied *S. aureus* phage Staph_1N either locally or both locally and orally for 24 days, which corresponds to one cycle of PT and 44 days, for the third PT cycle, respectively. High AAS levels were detected 2.5 months after treatment in patient no. 1 and on the 27th day of PT in patient no. 2 (K = 143.12 and K = 133.90, respectively). The AAS level decreased 1 year and 8 months after PT in patient no. 1 and 5 months after therapy in patient no. 2 (K = 22.18 and K = 31.95, respectively). In these two patients, clinical improvement was observed (discernible reduction in the intensity of some infection symptoms after completion of PT to a degree not observed before PT, when no treatment was used). Patients no. 3 and 4 used the *P. aeruginosa* phage Ps_2N or *E. coli* phage Ecol_L-4 either locally and orally or locally for 25 and 54 days, respectively. A medium level of AAS was detected 1 month after treatment in patient no. 3 and on the 53rd day of PT in patient no. 4 (K = 5.28 and K = 4.44, respectively). The level of AAS decreased 7 months after PT in patient no. 3 and 9 months after PT in patient no. 4 (K = 1.42 and K = 3.37, respectively). No response to treatment was observed in the patient using the *Pseudomonas* phage, whereas questionable clinical improvement could be seen in the patient using the *E. coli* phage.

Interestingly, we investigated earlier the humoral response by determining the production of anti-phage IgG, IgA and IgM antibodies, in sera of patients treated with phages, using the ELISA test [7]. Twenty patients used a lysate of *S. aureus* MS-1 phage cocktail orally and/or locally at the HIIET Phage Therapy Unit in Wrocław. The mean level of IgG, IgA, and IgM before phage therapy was 61.7 AU, 0.2 AU, and 11.8 AU, and during phage therapy it was 166.6 AU, 2.7 AU, and 205.9 AU. The increase during phage therapy was statistically significant for IgG, IgA, and IgM antibodies. A correlation was observed between higher IgG and IgM levels and the rate of phage inactivation K rate in sera of patients during phage therapy.

## 4. Discussion

The data presented here are of course very preliminary and incomplete. PT is still a rare form of treatment and published reports describing its effects frequently include single cases. Nevertheless, our data seem to provide relevant information for the further development of phage therapy. Thus, it appears that the level of anti-phage antibodies may drop after the completion of therapy while a significant decrease occurs after several months. Furthermore, the kinetics of antibody disappearance varies: in some cases, five months after termination of the therapy the antibody level was 36% of the immediate post-PT values while in another case that level was 76% compared with post-PT values. Finally, a significant reduction in very high levels may take more than a year (as seen in case 1).

As we have already pointed out, the clinical relevance of anti-phage antibodies is currently unclear and their presence in patients’ sera does not exclude successful therapy [5,6]. This finding has recently been confirmed by Dan et al. [11]. However, remote consequences are still unknown and it cannot be excluded that those antibodies and especially phage–antibody complexes may localize in different tissues and organs with subsequent injury [12]. Therefore, it is prudent to monitor patients receiving PT for anti-phage antibody levels and consider termination if the level rises significantly, keeping in mind our data indicating that a significant drop may take more than one year.

As recently pointed out by an editorial commentary in the Journal of Infectious Diseases, anti-phage immunity in patients receiving phage therapy can be considered as “fighting the wrong enemy” [13], since such humoral immunity can neutralize anti-bacterial effects of phages and therefore lead to the failure of PT. However, as pointed out by our group and confirmed by other authors, PT may be successful despite the development of phage-specific antibodies [11,14,15]. Evidently, our preliminary data on anti-phage antibodies after termination of phage therapy need to be expanded to shed more light on their immunopathologic and prognostic significance.

## 5. Conclusions

As already pointed out, our data are preliminary but suggest that the titer of phage-neutralizing antibodies falls after termination of therapy and the kinetics of this phenomenon may vary from patient to patient. Importantly, in some cases, a significant reduction in that titer may occur after one year. As suggested earlier, the appearance of those antibodies does not appear to directly relate to the clinical outcome of PT; however, this question requires further research. Further studies are needed to provide more data on anti-phage humoral response regulation during PT and its relevance for the immediate and remote outcomes of PT. Furthermore, the question of possible formation of phage–antibody complexes should also be explored, as currently no data are available on this important issue.

## Figures and Tables

**Figure 1 antibiotics-11-01358-f001:**
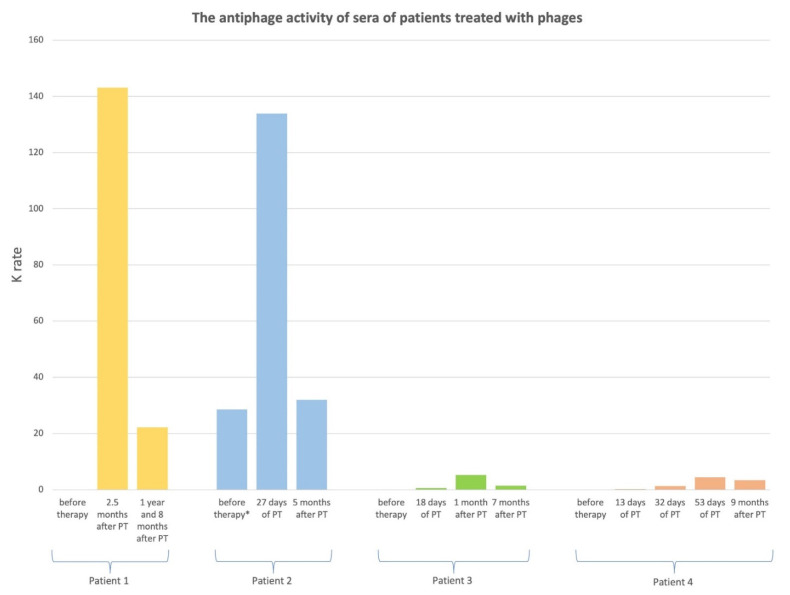
The antiphage activity of sera of patients treated with phages. Legend: PT—phage therapy; K: rate of phage inactivation; K < 5: low neutralization of phages; K = 5–18: medium neutralization of phages; K > 18: high neutralization of phages; * before the third PT cycle.

**Table 1 antibiotics-11-01358-t001:** Patients treated with phages.

Patient No.	Type of Infection	Route of Phage Administration	Target Pathogen	Phage Used in PT	Days of PT	Clinical Outcome of PT ^a^
1	Inflammation of the left hip joint	local	*S. aureus*	Staph_1N	24	C
2	Inflammation of the left ankle joint	The third PT cycle: local and oral	*S. aureus*	Staph_1N	44	C
3	Inflammation of the left hip	local and oral	*P.aeruginosa*	Ps_2N	25	F
4	Inflammation of the right calcaneus	local	*E. coli*	Ecol_L-4	54	D

Legend: PT—phage therapy; ^a^ Results A–C positive responses to PT; ^a^ Results D–G inadequate responses to PT.

## Data Availability

The data presented in this study are derived from personal patients’ medical records maintained at the Phage Therapy Unit of the Medical Centre as well as Bacteriophage Laboratory of the Institute of Immunology and Experimental Therapy Polish Academy of Sciences in Wrocław, Poland. Those data are not publicly available due to privacy and legal issues (The General Data Protection Regulation (EU) 2016/679 and Act on the rights of the patient and the Patient’s Rights Ombudsman from 6 November 2008).

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
