# Peer review of "Do Anti-Phage Antibodies Persist after Phage Therapy? A Preliminary Report"

_antibiotics, 2022, doi:10.3390/antibiotics11101358_

Round 1

Reviewer 1 Report

The work has been done well and it deserves to be published by this journal after addressing the below issues.

 The only major issues with this work are;

1.    What was the average K-rate for each patient at 30 days post phage therapy?

2.    Was it possible to detect other immunoglobins (such as IgA, IgG or IgM) in the patient’s sera besides the neutralizing antibodies?  

Reviewer 2 Report

In this study, Marzanna Łusiak-Szelachowska et al investigated phage-neutralizing serum antibodies in 4 patients with bone infection who were treated either locally or both locally and orally with phages. The study is well designed as a Preliminary Report. However, the authors should improve the presentation of the data in the manuscript. 

- The abstract needs to be re-written to highlight the findings in the paper.

- The anti-phage activity in the Table 1 would be better present a graph.

Reviewer 3 Report

The authors have taken up a novel topic for study, but there are several serious shortcomings.

1. The presentation doesn't follow the standard format.

2. Just taking up four cases to write a scientific article about a relatively new intervention makes the study very weak.

3. The abstract is too short and lacks clarity of content.

4. The introduction should have highlighted the issue in a compelling way  to the readers 

5.line 26 starting with" Our earlier data "should be rephrased with a snapshot of the earlier results 

6.The authors should stick to the standard format of writing the scientific article.Hence all the headings need to be rearranged.

7.Rationale of putting the "results" shortly after" introduction" before the "materials and methods" section.

8.No conclusions provided by the authors which defeats the whole purpose of the article.

9.References are very few in number which is not acceptable. 

10.Try including scholarly references from high impact journals.

11. The full identity  of the authors have been disclosed ,in stark contrast to ethical guidelines.

We look forward to hear from you at the earliest.

Thank you

Reviewer 4 Report

Overall, this is a well-written, precise, and clear manuscript. The introduction is relevant and theoretical in nature. The prior research findings are described in sufficient detail for readers to understand the current study rationale. The writers provide a systematic contribution to the scientific literature in this field of research. Overall, this is a good manuscript. It requires minor adjustments. Specific remarks are provided below.

Introduction:

In the 26th line give a brief description of phage therapy.

Please check all the references carefully.

Discussion:

In the 76th line “fighting wrong the enemy” please explain it.

Materials and method:

In the 92nd line explain HIIET.

In the 97th line write powers properly.

In the 101st line check the all units.

In the 110th line, what is P0?

Round 2

Reviewer 2 Report

The authors have addressed my concerns in the revision. 

Reviewer 3 Report

Thank you for taking the time to correct the issues pointed out.